# Low-Intensity Pulsed Ultrasound Effect on MIO-M1 Cell Viability: Setup Validation and Standing Waves Analysis

Irais Poblete-Naredo [1], Mario Ibrahin Gutierrez [2,*], Diana Estela Mendoza-Sánchez [3], Arturo Ortega [1], Arnulfo Albores [1], Josefina Gutiérrez-Martínez [4], Lorenzo Leija [5] and Arturo Vera [5]

1. Centro de Investigación y de Estudios Avanzados del IPN, Department of Toxicology, Mexico City 07360, Mexico; ipoblete@cinvestav.mx (I.P.-N.); arortega@cinvestav.mx (A.O.); aalbores@cinvestav.mx (A.A.)
2. CONACYT-Instituto Nacional de Rehabilitación LGII, Subdirección de Investigación Biotecnológica, División de Investigación en Ingeniería Médica, Mexico City 14389, Mexico
3. Unidad Profesional Interdisciplinaria de Biotecnología (UPIBI)-IPN, Mexico City 07738, Mexico; dmendozas2105tmp@alumnoguinda.mx
4. Instituto Nacional de Rehabilitación LGII, División de Investigación en Ingeniería Médica, Mexico City 14389, Mexico; jgutierrez@inr.gob.mx
5. Centro de Investigación y de Estudios Avanzados del IPN, Department of Electrical Engineering-Bioelectronics, Mexico City 07360, Mexico; lleija@cinvestav.mx (L.L.); arvera@cinvestav.mx (A.V.)
* Correspondence: migutierrezve@conacyt.mx; Tel.: +52-555-999-1000

**Abstract:** Low-intensity pulsed ultrasound (LIPUS) has been proposed for novel therapies still under study, where similar parameters and protocols have been used for producing opposite effects that range from increasing cell viability to provoking cell death. Those divergent outcomes make the generalization of expected effects difficult for cell models not yet studied. This paper presents the effect of LIPUS on the viability of the MIO-M1 cell line for two well-established setups and different protocols; the acoustic intensities, duty factors, and treatment duration were varied. Measurements and models for acoustic and thermal analysis are included for proposing a solution to improve the reproducibility of this kind of experiments. Results indicate that MIO-M1 viability is less affected for the cells treated through a dish that is partially immersed in water; in these conditions, the cells neither show detrimental nor proliferative effects at intensities lower than 0.4 W/cm$^2$ at 20% duty factor. However, cell viability was reduced when LIPUS was followed by cell subculturing. Treating the cells through a gel, with the culture dish placed on the transducer, increases cell mortality by the production of standing waves and mixed vibration-acoustical effects. Using the water-based setup with a 1° dish inclination reduces the effects of standing waves.

**Keywords:** LIPUS; MIO-M1; cell viability; standing waves analysis; ultrasonic in vitro heating; finite element analysis

## 1. Introduction

Ultrasound has been proposed for an extensive variety of therapies with different modalities and intensities for producing either thermal or non-thermal effects. The low-intensity pulsed ultrasound (LIPUS) is a type of non-thermal ultrasound modality with intensities lower than 3 W/cm$^2$ applied in a pulsed regime that has been proposed for multiple medical applications [1–7]. Interestingly, those therapeutic applications range from tissue healing and cell survival stimulation to cell death induction (cancer therapy), in which comparable acoustic treatment conditions are used for producing opposite effects. For instance, ultrasonic parameters for cell apoptosis induction in cancer therapy [1,2,8,9] are quite similar to those used for induction of increased survival under basal [3,10] and stressing conditions [6]. These discrepant outcomes make the protocol designing for cell models that are not yet studied difficult.

Moreover, there are diverse cell treatment setups that have been used with miscellaneous results. One of the most used setups for in vitro cell stimulation places the transducer below the cell culture plate with ultrasonic gel as a coupling media [1,11]. Using this setup permits a mechanically stable configuration, with the disadvantage of increased standing waves in the culture media and, consequently, a high variability of results [12]. In another configuration, the culture plate is situated on the surface of a water tank above the planar transducer, which is separated from the dish at a certain distance [2,10]; this arrangement has shown to have reduced standing waves and give more reproducible results [12]. However, other experimental configurations in which the cells are placed in a closed chamber immersed in a water tank [13,14], or where the transducer is inserted into the cell culture media for cell stimulation from above [7,15] have been scarcely used. Acoustic pressure measurements complemented with computational models suggest that those discrepancies in treatment setups originate a high variability of results and a low level of comparison between studies [12].

Differences in stimulation protocols increase the variability in outcomes observed for in vitro stimulated cell cultures. Some authors made use of a single ultrasonic treatment that ranges from seconds to 20 min in duration [1,7,8,10] while others fractionate the ultrasonic dose in three short treatments separated by a lapse of an ultrasonic recess [4,16]. Moreover, the ultrasound stimulation can be applied once a day during two or more days [11,17–19], or every other day [14]. Furthermore, the ultrasound can be applied using planar or focused transducers with different acoustic intensities, duty factors (DF), and pulse repetition frequencies (PRF) that increase the number of studied variables [20].

Therefore, the development of an in vitro cell stimulation protocol with LIPUS should account for different types of variables; for instance, acoustic intensity, DF, PRF, setup configuration, time of exposure per session, and number of sessions per day/week. Those parameters can be tested in terms of the viability of each cell type to analyze according to the expected ultrasonic effects. In this paper, the viability analysis of a retinal glial cell line, MIO-M1, treated with LIPUS is presented. LIPUS was applied from the bottom of the culture dish using two setups commonly used for ultrasonic treatments of adherent cells. The viability was determined for different combinations of acoustic intensities, DFs and treatment durations. For validating the setup, the ultrasound transmission through the culture dish was measured; a finite element model of ultrasound and heat propagation in the culture well was analyzed. The reduction of standing waves using small relative inclinations between the transducer and the dish is proposed.

## 2. Materials and Methods

### 2.1. Cell Culture

MIO-M1 cells, a human cell line with retinal Müller glial cell characteristics, were kindly provided by Prof. Ana María López-Colomé (UNAM, Mexico City, Mexico). Cells were cultured in DMEM-high glucose medium supplemented with 10% fetal bovine serum (FBS) and a mixture of antibiotics and antimycotics (GIBCO, Life Technologies, Carlsbad, CA, USA) in 35 mm polystyrene culture dishes (Corning Inc., Corning, NY, USA). Cells were incubated at 37 °C in a 5% $CO_2$ atmosphere in a humidified incubator chamber until 80% confluence was achieved and LIPUS stimulation was performed.

### 2.2. Experimental Setup for LIPUS Treatments

LIPUS was applied from the bottom of the culture dish (Figure 1a) using an ultrasound transducer of 1 MHz and 10 $cm^2$ of nominal radiating area (Model 7310, Mettler Electronics Corp., Anaheim, CA, USA). The transducer was driven in pulsed mode with a square-modulated sine-wave by using a custom-made RF amplifier with 10 Hz of PRF and adjustable DF. The spatial average acoustic intensity was determined using a radiation force balance (UPM-DT-100N, Ohmic Instruments Company, St Charles, MO, USA). The culture dish was placed directly on the planar transducer using acoustic gel as a coupling media (well-on-transducer setup) or it was sealed with a plastic wrap (to prevent water infiltra-

tion) and placed with the bottom part into the water surface (well-on-water-surface setup) by using a custom-made metallic holder [12]. In the second case, the distance between the transducer and the dish was 1.0 cm, and the depth of the culture media was 2.0 mm (2 mL). The holder did not interfere with the ultrasound radiation. Water temperature was adjusted at 37 °C; it was not closed-loop controlled to avert extra vibration. The variation of water temperature during the 1–10 min experiments was considered negligible.

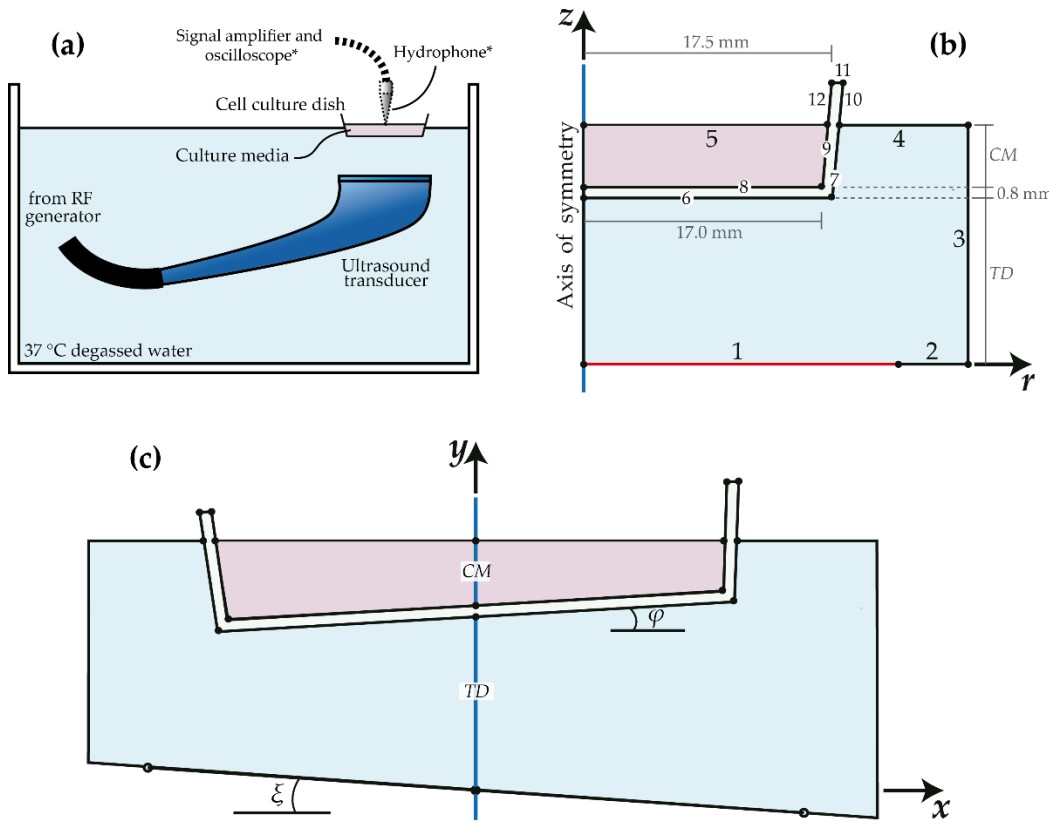

**Figure 1.** (**a**) Well-on-water-surface setup for ultrasound treatments. The components marked with (*) were used during the ultrasound transmission characterization. (**b**) 2D axisymmetric geometry for acoustic and thermal simulations. (**c**) 2D geometry for standing waves analysis (inclinations $\xi$ and $\varphi$ are not scaled). *CM* = depth of culture medium. *TD* = transducer to dish distance.

### 2.3. Biological Assays for Cell Viability

LIPUS treatments consisted of different intensities and exposure times. Five spatial average acoustic intensities of 0.2, 0.4, 0.8, 1.2 and 1.6 W/cm$^2$ with three DFs of 10%, 20% and 50% during 1 min were applied; these experiments were performed for both setups. The rest of the experiments were performed with the water tank setup (Figure 1a) using different treatment duration of 1–10 min at 0.4 W/cm$^2$ and 20% DF (spatial-average temporal-average acoustic intensity, $I_{SATA}$ = 0.08 W/cm$^2$). For all the above experimental conditions, cells were placed back in the incubation chamber after LIPUS stimulation and incubated for 24 h. In the experiments where cell adhesion and proliferation capability were assessed, cells were stimulated at the indicated periods and LIPUS intensities, and further passaged at a 1:10 dilution factor in 24 multi-well plates (by duplicate or quadruplicate). After ultrasonic stimulation, cells were incubated for 24 h or 48 h in an incubation chamber. One hour before the incubation period was over, 3-(4,5-Dimethyl-2-thiazolyl)-2,5-diphenyl-2H-tetrazolium bromide (MTT, Sigma-Aldrich, Saint Louis, MO, USA) reactant was added to the cell culture at 0.5 mg/mL final concentration. Lastly, the cell medium was removed, the precipitated formazan salts were dissolved in dimethyl sulfoxide, and absorbance was measured in a spectrophotometer (Infinite M200, Tecan, Switzerland) at 570 nm excitation

and 660 nm reference wavelength. Cell viability was calculated by the subtraction of the absorbance value at 660 nm to the absorbance at 570 nm (adjusted absorbance, AA) and was presented as % of control that is the result of (AA of the experimental condition/AA of control condition) $\times$ 100.

MTT results represent the $\pm$ standard error of the mean ($\pm$SEM) of three independent experiments at each condition, unless otherwise stated. Data were tested for normal distribution with the Shapiro-Wilk normality test. Assays with normal distribution were analyzed with one-way ANOVA and further differences against the control were established with Dunnett's multiple comparison test. Data that did not pass the normality test were analyzed with the Kruskal–Wallis test and significant differences against the control group were established with Dunn's multiple comparison test. Statistical differences were set at $p < 0.05$. Statistical analysis was performed with GraphPad Prism v7 (GraphPad, San Diego, CA, USA).

### 2.4. Determination of Ultrasound Transmission

The transmission of ultrasound through the culture dish was determined using the setup of Figure 1a (*). The culture medium was substituted with distilled degassed water, based on previous data that indicates both water and medium share similar acoustic parameters [12]. A wide-band needle hydrophone HNP-1000 (Onda Corporation, Sunnyvale, CA, USA) with an aperture of 0.4 mm and sensitivity of 220 nV/Pa was used to measure the acoustic pressure at the water surface; the hydrophone was placed into the water at 5 mm from the bottom of the well. The hydrophone was moved with an automatic 3D scan system (Scan 340, Onda Corporation, Sunnyvale, CA, USA) following the well's diameter along the X-axis from $x = -15$ mm to $x = -15$ mm at steps of 0.3 mm; this small step corresponds to 1/5 of wavelength in water at 1 MHz. The geometrical center of the culture dish was used as the zero position ($x = 0$, $y = 0$) of the coordinate system. The hydrophone was not moved along the Z-axis. The transducer was driven with 10 $V_{p-p}$ tone-burst of ten sine-cycles using a wave generator (Array 3400, Array Electronic Co., Taiwan, China) to emulate a nearly continuous emission that produces a quasi-stationary radiation pattern. The produced $V_{p-p}$ at the hydrophone was 20 dB amplified (AH-2010, Onda Corporation, Sunnyvale, CA, USA), and recorded in a PC through a software-controlled oscilloscope (TDS2042B, Tektronix, Beaverton, OR, USA).

### 2.5. Models Equations and Conditions

The ultrasound transmission under experimental conditions was modeled to determine the effect of the culture dish and its relative inclination in both the radiation pattern and the total delivered energy. The heat in the system produced by ultrasound absorption was also analyzed. Both models were developed with the finite element method (FEM) using COMSOL Multiphysics 5.5 (COMSOL AB, Stockholm, Sweden) in a PC-workstation with 8-core 3 GHz microprocessor and 64 RAM (Dell, Round Rock, TX, USA). When required, modeling results were post-processed using MATLAB 2018a (MathWorks, Natick, MA, USA).

The ultrasound transmitted pattern and the heat produced in the dish were modeled assuming a 2D axisymmetric geometry (Figure 1b) based on the symmetry of the culture dish and the measured field [21]. The transducer radiation was set on boundary 1; boundaries 2 and 3 were configured to have the same acoustic impedance of water to reduce wave reflections at the walls; boundaries 4, 5, 10, 11 and 12 were set as hard boundaries (zero particle velocity) to represent the large differences of the acoustic impedances between the respective media (water or polystyrene) and the air; interior boundaries had a continuity condition. The pressure radiating at boundary 1 had a radial pattern of a simple supported Bessel function [21] given by

$$p(r) = P_0 \left[ C_1 J_0 \left( \beta_{2N-1} \frac{r}{a} \right) + C_2 \right] \cdot \sum_{m=1}^{4} \left[ 1 - \left( \frac{r}{a} \right)^{2m} \right] \tag{1}$$

where $r$ is the radial coordinate, $a$ is the transducer radius (m), $J_0$ is a Bessel function of the first kind of order zero, and $P_0$ is the spatial-averaged pressure amplitude at the transducer surface (Pa), which depends on the acoustic intensity [22]; $C_1 = C_2 = 0.5$, $\beta = 33.776$, $m = 4$ and $N = 6$, which are detailed in [21].

Under harmonic conditions, standing waves can be produced at distances that are multiples of half-wavelength of the ultrasound in the transmitting media [12]. For studying these resonances, an additional 2D geometry (no-axisymmetric) was created (Figure 1c); this second geometry was based on Figure 1b with the addition of an identical mirrored section. Boundary conditions are those mentioned for Figure 1b, since the boundaries are correspondent; the radiating boundary was also set using (1) by substituting $r$ with $|x|$.

Using this second geometry, parametric sweeps at variable transducer-to-dish (*TD*) distances and depths of culture medium (*CM*) were carried out; the latter distance was varied from 9.0 to 12.0 mm while the former from 1.0 to 3.5 mm, both with steps of 0.1 mm. Additionally, it was studied the suppression of the standing waves produced by relative inclinations between the transducer and the dish during experiments; for this, we hypothesized that small inclinations could improve the repeatability of the results. The dish subdomain and the transducer boundary were rotated as shown in Figure 1c (sketched inclinations are not scaled). Therefore, an additional parametric sweep, combined with the two previously described sets, was computed by varying both angles, $\zeta$ and $\varphi$, from $-2°$ to $2°$ at steps of $0.5°$. The amplitude of the acoustic pressure at the surface of the culture medium was averaged to determine the acoustic pressure transmitted at each parameter combination. The mesh in both geometries consisted of 15 quadrangular elements per wavelength, which represents 85,000 elements for the 2D model. Mesh convergence was verified by increasing the mesh resolution with less than 0.019% of the variation of the pressure amplitude.

The radiating pattern at the dish was modeled with a time-dependent simulation using the 2D axisymmetric geometry and a tone-burst of ten sine-cycles with the radial distribution of (1). The peak-to-peak pressure at the water surface was obtained. The acoustic properties of the subdomains of this model are shown in Table 1. The time dependent ultrasound propagation was determined with FEM by solving the acoustic wave Equation (2) given by

$$\nabla^2 p - \frac{1}{c^2}\frac{\partial^2 p}{\partial t^2} = 0, \tag{2}$$

where $p$ is the acoustic pressure (Pa), and $c$ is the speed of sound in the media (m s$^{-1}$).

**Table 1.** Acoustic and thermal properties of materials used in simulations [12,23–26].

| Material | Acoustic Attenuation [Np m$^{-1}$] | Speed of Sound [m s$^{-1}$] | Heat Capacity [J kg$^{-1}$ K$^{-1}$] | Thermal Conductivity [W m$^{-1}$ K$^{-1}$)] | Density [kg m$^{-3}$] |
|---|---|---|---|---|---|
| Water | 0.23 | 1523 | 4183 | 0.610 | 993 |
| DMEM | 0.23 | 1543 | 4183 | 0.610 | 998 |
| Polystyrene | 4.03 | 2350 | 1195 | 0.115 | 1053 |

The acoustic field was also modeled under harmonic conditions to determine the effect of standing waves for different setup inclinations. Although the ultrasound transducer was driven in pulsed regime, we can assume acoustic harmonic conditions based on the large number of sine-cycles into each pulse, e.g., 10,000 sine-cycles in 1-MHz tone-burst with 10% DF at 10 Hz PRF. The amplitude of the radiated pressure of boundary 1 was $P_0 = 155$ kPa, which is equivalent to 1.6 W/cm$^2$ at 50% DF for this transducer [22]. The acoustic field was determined based on the harmonic acoustic wave equation for attenuating media [21].

The heat produced in the culture dish was studied for the largest intensity used in the experiments (1.6 W/cm$^2$ at 50% DF). From Figure 1b, *TD* = 1.0 cm and *CM* = 2.0 mm. Materials' properties for acoustic and thermal models are shown in Table 1. The temperature in

the system was modeled using the parameters and conditions described in [22]. Boundaries 1 to 3 of Figure 1b were set with a constant temperature of 37 °C; boundaries 4, 5, 10, 11 and 12 were set with zero heat transfer assuming null heat losses by both radiation and thermal conduction to the air. Interior boundaries were set with a continuity condition. Natural convective heat transfer was assumed small with a velocity component of 0.1 mm/s in the *z*-direction [27]. Mesh was kept the same as in the acoustic models. The time dependent heating simulation was computed from 0 to 300 s (5 min) at steps of 1 s. The acoustic attenuation was assumed produced by only absorption (zero scattering).

## 3. Results

### 3.1. LIPUS Effect on MIO-M1 Cell Viability

In vitro experiments were performed to evaluate the acoustic effect on the viability of the cell line MIO-M1. The effect of increasing acoustic intensities and DFs at 1 min treatment length on MIO-M1 viability, 24 h post-treatment, is reported in Figure 2a,b. In the first case (Figure 2a), treatments were performed with the well-on-transducer setup (as detailed in Material and Methods section). We observe that the increment in the acoustic intensity at 10% DF has negligible effects on cell viability. In contrast, at 20% and 50% DF, there are significant effects on the viability of the cells treated at increasing LIPUS intensities. The treatment at 1.6 W/cm$^2$ and 20% DF produced statistically significant effects on cell viability with only 23.8% of viable cells remaining after stimulation ($^{++}$ $p < 0.01$, post-hoc Dunnett's test). The increase in the DF to 50% reduces the cell's resistance to increasing acoustic intensities. Substantial cell death is observed at 0.8, 1.2 and 1.6 W/cm$^2$ at 50% DF, where the percent of cell death was 75.0%, 94.5% y 93.7%, respectively (*** $p < 0.001$, post-hoc Dunnett's test).

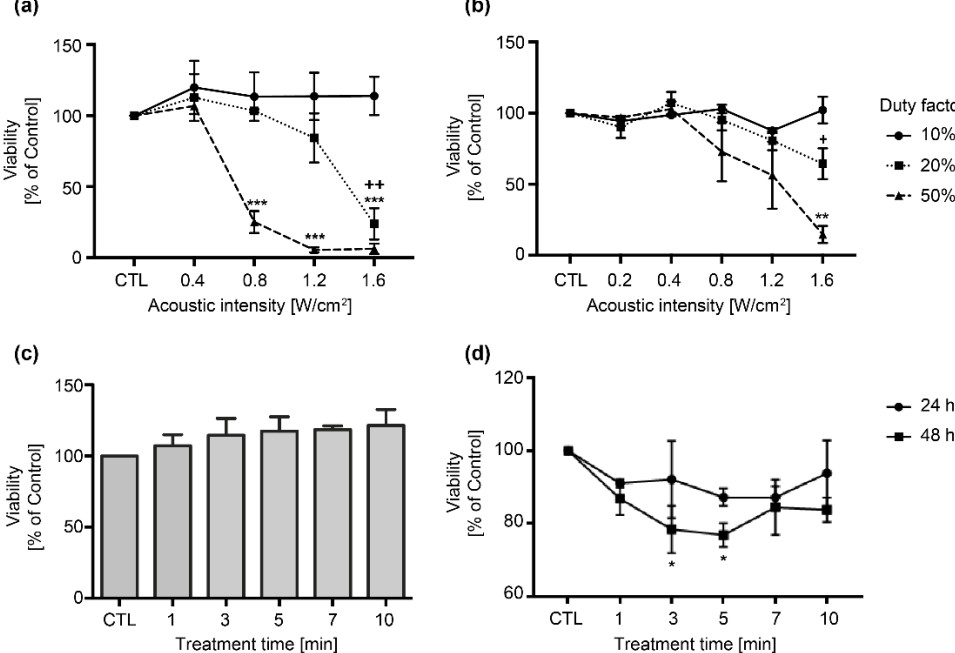

**Figure 2.** (**a**,**b**) Viability of MIO-M1 cells at different acoustic intensities and duty factors (10%, 20% and 50%) for 1 min of treatment when using (**a**) well-on-transducer setup ($^{++}$ $p < 0.01$, CTL vs. 1.6 W/cm$^2$, 20% DF; *** $p < 0.001$, CTL vs. 0.8–1.6 W/cm$^2$, 50% DF; post-hoc Dunnett's test), and (**b**) well-on-water-surface setup ($^{+}$ $p < 0.05$, post-hoc Dunn's test; ** $p < 0.01$, post-hoc Dunnett's test). (**c**,**d**) Viability of MIO-M1 for different treatment durations with 0.4 W/cm$^2$ at 20% DF evaluated with the water-based setup (**c**) 24 h after treatment, (**d**) 24 h and 48 h post-LIPUS treatment and sub-culturing (* $p < 0.05$, post-hoc Dunnett's test). All experiments were performed in triplicate, except for the viability curves at 10% DF through gel (**a**) and 24 h after LIPUS treatment and cell passaging (**d**), which were performed by duplicate.

When the setup is changed to a water immersion bath set at 37 °C (Figure 2b), the acoustic effect on the cell mortality is markedly reduced. Cell death was significant only at 1.6 W/cm$^2$ at 20% DF ($^+$ $p < 0.05$, post-hoc Dunn's test) or 50% DF (** $p < 0.01$, post-hoc Dunnett's test), with reductions on cell viability to 64.4% and 14.7%, respectively.

Due to the reduced acoustic effects in the cell viability using a water-based setup, we further evaluate the effect of increasing time exposure by using a fixed acoustic intensity at 0.4 W/cm$^2$ and 20% DF. Figure 2c illustrates that the MIO-M1 cell viability is not affected with one to ten minutes of ultrasonic treatment under these acoustic conditions. Average water temperature, set at 37 °C at the beginning of the experiment and measured after 10-min treatments, at three different days, was 36.5 °C $\pm$ 0.2 °C, which should not produce any effect in cell viability

In order to determine the proliferative capacity of cells stimulated with LIPUS, we exposed MIO-M1 cells to the same acoustic intensity of 0.4 W/cm$^2$ at 20% DF with also varying treatment lengths. Further, cells were trypsinized, diluted (1:10), and seeded in 24-well plates (Figure 2d). We observe that 24 h after cell exposure and passage, cells seem to maintain their ability to adhere and proliferate as observed in the non-treated condition (CTL). Notwithstanding, cell viability decreases to around 80% at all treatment times 48 h after LIPUS stimulation and passage. This reduction is significant at 3 min and 5 min treatments (* $p < 0.05$, post-hoc Dunnett's test), where viability is decreased to 78% and 77%, respectively.

### 3.2. Analysis of the Delivered Acoustic Field

The acoustic field in the culture dish was measured and modeled. Measurements were carried out with a wide-band hydrophone when the transducer was emitting a 10 sine-cycles tone-burst. Figure 3a shows the measured radiation pattern of ultrasound transmitted through the culture dish compared with the no-dish condition. The measured pattern for no-dish condition corresponds to a Bessel-like profile characteristic in the Fresnel-zone of no-backed planar transducers [21]. The acoustic field measured through the dish shows clear differences in shape from the no-dish condition. The field at the central zone of the culture dish presents a slight energy concentration of 7.8% (with a relative reduction at other more external zones) that could be influenced by wave reflections at the surface of the water that was agitated by the acoustic radiation force; this vibration was perceptible even with bare eyes and it was more intense at the center of the culture dish. This behavior is not replicated by the models since water agitation is not included.

The modeled acoustic patterns at the surface of the water for the dish and no-dish conditions are shown in Figure 3b. The radiation profiles between the two conditions are similar at the central zone of the well with a slight variation at the border of the dish that can be also noticed in the measurements. The acoustic attenuation of the dish combined with 94% of the transmission coefficient at the two interfaces water–polystyrene [28] produces an average amplitude reduction of 9% that is congruent with measurements. The measured attenuation of the dish was 4.01 Np/m congruent with Table 1. Although the shape of the real field measured through the dish is modified by water vibration, the average pressure corresponds to the expected one, as verified by the models, with a small effect of attenuation and transmission losses.

Standing waves produced by both parallel and inclined surfaces are analyzed for our setup. Parametric simulations varying *TD* and *CM* were carried out to determine the combinations of those distances that produce resonant fields. The average pressures at the surface for each parameter combination are shown in the 2D color plots of Figure 3c,d for the models with parallel and inclined boundaries, respectively. From the pattern of Figure 3c, standing waves increase the average pressure at the surface following the expected tendency, i.e., resonances are produced at distances that are multiples of half-wavelength of ultrasound in media. The observed pattern forms circle-like regions where the acoustic pressure is not increased, with a small central zone at half pressure amplitude that was produced by anti-resonant destructive interference. When the dish is inclined relatively to the transducer,

$\xi = 2°$ and $\varphi = -1°$, the resonances are importantly attenuated (Figure 3d). This reduction is mostly true when the dish and the transducer are individually rotated in the same direction (notice the opposite sign of the angles) to have at least 1° of relative inclination. Rotations at opposite directions (as the drawing of Figure 1c) produced mixed results showing either reduction or creation of standing waves at certain angles combinations (data not shown).

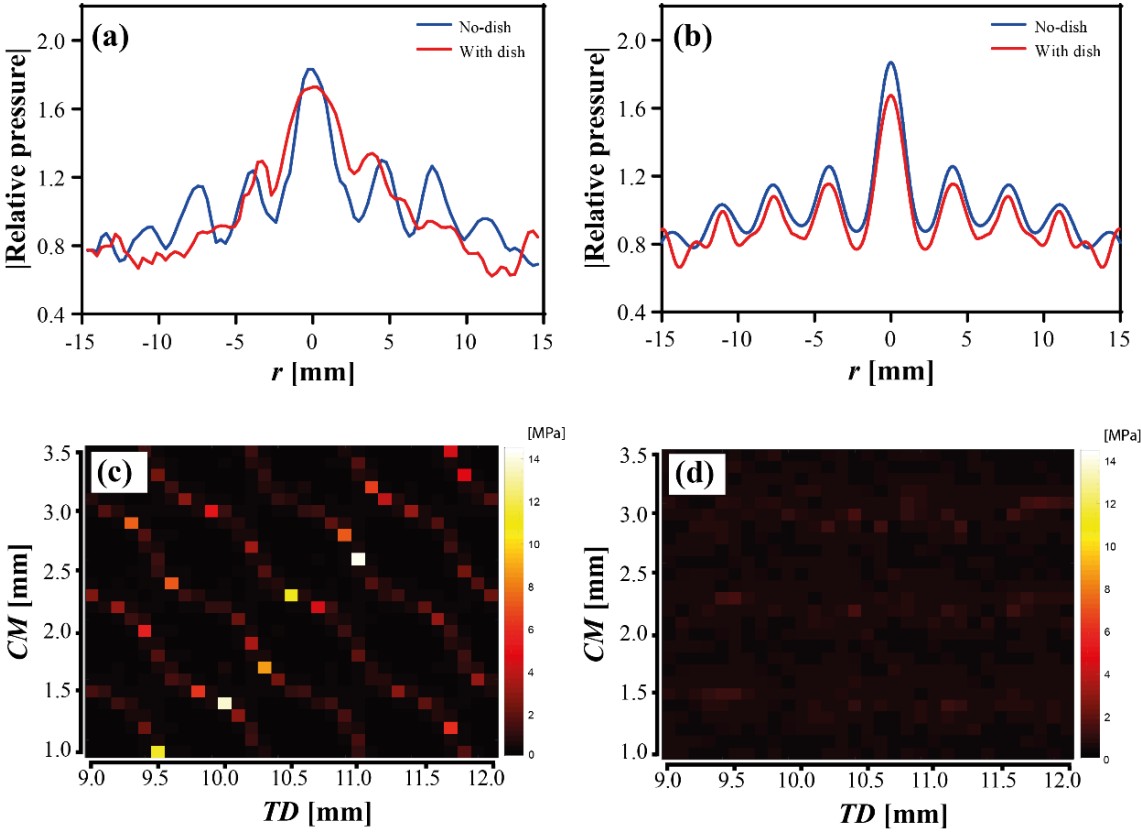

**Figure 3.** (**a**) Measured ultrasound pattern through the culture dish compared with the no-dish condition. (**b**) Model of transmitted ultrasound pattern through the dish and no-dish condition using a time-dependent simulation of 10 sine-cycles. (**c**) Resonances for different distances between the transducer and the culture dish (*TD*) versus the depth of culture medium (*CM*) with the dish and transducer surfaces perfectly parallel. (**d**) Resonances for the same distances of (**c**) with $\xi = 2°$ and $\varphi = -1°$. For (**a**) and (**b**), the acoustic pressure is relative to the average pressure in free-space at the transducer surface in the effective radiating radius [22].

### 3.3. Heat Generation in the Culture Dish

The heat generated by LIPUS was modeled according to the experimental conditions. The thermal modeling approach using a two-step solution was previously validated by our group [22]. Figure 4 shows the temperature distribution in the culture dish for the most extreme experimental condition of this work, which is 1.6 W/cm$^2$ at 50% DF. The maximum temperature at the center of the culture dish was 37.76 °C that should be insufficient to modify cell viability; other zones of the well have a more uniform temperature below 37.50 °C. Moreover, forced convection by water agitation was not included in the model, which could uniformize the temperature in the well and reduce hot-spots.

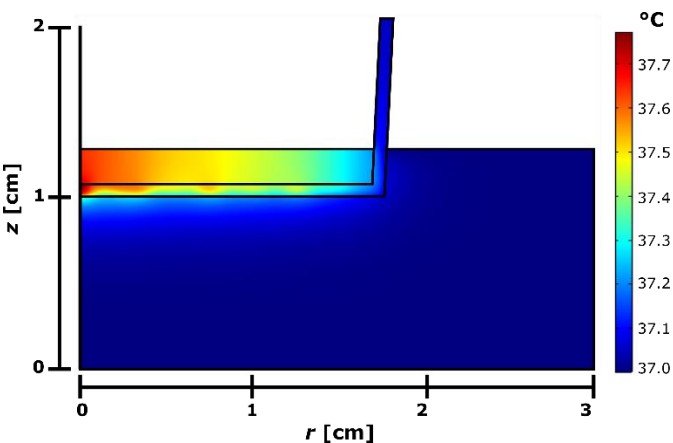

**Figure 4.** 2D axisymmetric modeled temperature for the experiments with ultrasound at 1.6 W/cm$^2$ at 50% DF after 5 min. A maximum temperature increment of 0.76 °C should not affect cell viability. This figure represents the result with the most intense experimental condition used in this work. Agitation produced by the acoustic force, not considered here, would also uniformize the heat in experiments, which would reduce the average temperature.

## 4. Discussion

Human retinal cells MIO-M1 were radiated with LIPUS for evaluating cell viability after treatments. Ultrasound was applied from the bottom of the culture dish using two common experimental setups. Results using both setups were compared for determining the approach that presents less cell death induction. The setup with the culture well partially immersed in water was validated with acoustic field measurements and finite element models. Cell viability was determined for different acoustic intensities, DFs, and treatment durations. PRF was set at 10 Hz due to high viability and low cell death observed with this parameter as reported by [20].

### 4.1. Cell Viability after LIPUS with Different Setups

Diverse experimental setups for in vitro cell exposure with LIPUS have been described in the literature with the support of models and hydrophone measurements [12]. Due to their low cost, practicality and application to adherent cells, we evaluated two experimental setups: the well-on-transducer and the well-on-water-surface configurations. We observed that LIPUS applied with the gel-based setup presents greater detrimental effects on cell viability than LIPUS application with the water-based setup at similar acoustic conditions (Figure 2a,b). Although the acoustic energy applied in both cases was the same, the viability of cells treated with LIPUS for the first case decreased at acoustic intensities larger than 0.4 W/cm$^2$ at 50% DF ($p < 0.001$), while the cells treated using the well-on-water-surface setup tolerated larger intensities with a significant viability decrease only at 1.6 W/cm$^2$ at 50% DF ($p < 0.01$). Similar results were observed in the viability decline of the hepatocarcinoma cell line, SMMC-7721, at 1 min stimulation with 0.5 W/cm$^2$ (0.25% DF) or higher LIPUS intensities using the well-on-transducer setup [1]. In contrast, HepG2 and 3T3 cells exhibited significant viability reductions only at high LIPUS intensities of 1.2 W/cm$^2$ at 50% DF when stimulated in a well-on-water-surface configuration [10].

According to [12], both studied setups present the disadvantage that variations in the height of the culture medium, which determines the distance between the cell monolayer and the air interface, impact the amplitude of acoustic pressure. Nevertheless, the well-on-water-surface setup tolerates larger variations in media volume than the well-on-transducer configuration; for instance, the former presents a 10% reduction in pressure amplitude when the height deviates 0.46 mm, while the latter displays the same reduction for variations of 0.02 mm in media's height. This particularity of the well-on-water-surface setup should permit to have more robust results among repetitions.

However, two additional hypotheses could complement the explanation of the differences in the cell viability between gel-based or water-based setups, which are not mutually excluding. First, when using the well-on-transducer setup, an extra mechanical vibration from the transducer could be transmitted by direct contact to the culture dish. This additional vibrational wave may increase the overall cellular effect when combined with the pure ultrasound propagation; this combination of phenomena was already proposed to be a significant contributor for the whole ultrasonic effects in other applications [29]. Second, the resonance effects may be increased when the transducer is directly coupled to the dish, probably due to the parallelism of the multiple interfaces that enhance the possibility of producing standing waves in media [12]. Both, mechanical vibration and resonance effects may be the leading factors that cause the instant cell detachment observed with the ultrasound stimulation at 0.8 to 1.6 W/cm$^2$ and 50% DF (data not shown). In contrast, as both phenomena are reduced in the well-on-water-surface setup, there is a marked reduction in the instantaneous cell lysis, but instead, cells undergo delayed cell death processes which comprise early and secondary apoptosis [1,8]. The manner the well-on-transducer setup impacts cell viability by cell detachment may account for the low data variability observed, in comparison with the large variation computed in the well-on-water-surface setup, which generates medium and long-term cellular effects. These results are particularly important since LIPUS has been applied indistinctly with any of the studied setups, and the biological impact associated with the use of either, as far as we have knowledge, has not been previously discussed.

### 4.2. Effect of Increasing LIPUS Intensity and Exposure Time

The effect of LIPUS has been addressed in different cell types with varying effects [1,6,8]. In this paper, MIO-M1 response to LIPUS appeared not to be detrimental for cell viability in the range of 0.2–1.2 W/cm$^2$ acoustic intensities at 10%–20% DF when a water immersion setup was used (Figure 1b). Nevertheless, we were unable to detect an increased cell proliferation effect at these experimental conditions, as reported in HepG2 and 3T3 cells. While Yang et al. [10] registered a statistically significant proliferation increase in cell cultures treated for 1 min with 0.2 and 0.43 W/cm$^2$ at 50% DF and 100 Hz PRF, we did not detect cell proliferation at almost the same ultrasound conditions in MIO-M1 cells with 10 Hz PRF.

The cell viability dependence on PRF has been addressed by [20]. They observed that a high percentage of viable cells is achieved in treatments with 5–100 Hz PRF and further decreases at lower PRF, probably due to ultrasonic streaming. Despite they did not show significant viability differences between 5 and 100 Hz PRFs, we suggest that streaming may impact the ultrasound mediated proliferative effects. Nevertheless, further studies are suggested to determine PRF influence in LIPUS cell proliferation capacity.

Moreover, differences in LIPUS biological effects between cell populations cannot be discarded. Other authors did not observe changes in the cell proliferation rate in different cell lines [5] or primary cell cultures [11] exposed to a repetitive daily-based LIPUS regime. On the other hand, additional extracellular components could contribute indirectly to the LIPUS proliferative stimulation by releasing paracrine signaling factors [11].

### 4.3. Cell Viability 24 h and 48 h after LIPUS

Further, the effect of increased treatment duration was evaluated for a fixed intensity of 0.4 W/cm$^2$ at 20% DF. This intensity was chosen since not visibly changes were observed at 1 min treatment at any DF. Figure 2c shows no effects 24 h after increasing treatment times (1, 3, 5, 7 and 10 min). Again, we were unable to detect proliferative nor detrimental effects on MIO-M1 viability.

In order to better characterize the LIPUS effects on MIO-M1 viability, we diluted (1:10) the cell population by trypsinization and sub-culturing after LIPUS treatment. This way we would be able to distinguish if cells maintained their ability to proliferate and if they presented delayed LIPUS effects for longer incubation periods after exposure. By means of

this protocol, we observed (Figure 2d) a significant decline in cell viability at 3 min and 5 min treatment after 48 h treatment and passage. In fact, that trend was already visible after 24 h. These results are consistent with [8]. They observed that at 0.3 W/cm$^2$ and 10% DF, there is minimal instant cell lysis, but cell death starts increasing 6–24 h post-treatment. The values they observed of total cell death at 24 h incubation post-treatment were highly variable, as we also observed (Figure 2b), and they proposed that cell mortality was the product of secondary cell necrosis and early apoptosis. We suggest, therefore, that the decrease in cell viability in our model at 48 h is a product of the long-term effect of LIPUS to induce secondary cell necrosis.

*4.4. Models and Measurements for Setup Validation*

The acoustic field transmitted through the dish and the heat produced in the system were modeled under the specified experimental conditions. The measured radiation profile at the surface of the culture medium (or water in experiments) was congruent in amplitude with the modeled field (Figure 3a,b). For the no-dish condition, the modeled distribution adequately follows the measured profile, with minor variations. The relative amplitudes are also congruent with overall differences of 9% caused by attenuation and transmission losses, which confirms the emitted radiation of (1) is valid for this transducer. However, the acoustic pattern measured through the dish differs from the modeled field. Although the average amplitude is accordingly reduced by the effect of attenuation and transmission losses, the local pressure amplitude is increased at the central region ($r < 5$ cm) and reduced at more external zones for $r > 5$ cm. The modeled field does not show that deformation when the dish is included. The deformation at the center of the field in experiments could be produced by the water movement at the surface provoked by the acoustic force. The displacement of the water, more intense at the center of the well, modifies the shape of the water-air interface, which increases the concentration of the field at that zone. It was not easy to replicate this behavior in the models to demonstrate this hypothesis; however, since average amplitudes were correctly reproduced, we considered that this demonstration was not required. Moreover, using an acoustic absorbent cap in a full filled culture well could be an alternative to reduce standing waves and avoid water agitation at the surface [30,31]; this alternative can be studied in future work.

When varying the distances between the transducer and the dish (*TD*) and the depth of the culture medium (*CM*), some resonant behaviors were found. Standing waves are produced at distances that are multiples of half-wavelength of ultrasound in the transmission medium [12]. Figure 3c shows the effect of standing waves in the averaged pressure amplitudes at the surface of the culture medium when the bottom of the dish and the surface of the transducer were perfectly parallel. As predicted, resonances were generated each half-wavelength of both culture medium (vertical trend) and water (horizontal trend), which creates circle-like regions of resonant amplitudes with no-resonant interior zones. At the very center of the dark zones, single points of anti-resonant conditions are produced by destructive interference which reduces the average pressure by half of the emitted pressure. The production of standing waves is regular and predictable when the surfaces are perfectly parallel. Considering the ultrasound wavelength in water is around 1.5 mm (temperature dependent), there is a resonant point each 0.75 mm that would create unpredictable and inconsistent results during cell treatments. In order to have reproducible results using this setup, it is expected to work in the dark circular regions outside the center.

The standing waves were reduced when the model included the inclinations of the dish and the transducer. Figure 3d shows a representative result when the dish was inclined 1° relative to the transducer surface, and the culture medium was inclined another 1° relative to the bottom of the culture dish. The average amplitudes of the inclined setup were smoother with less resonant zones that help us finding adequate regions to work without standing waves. Although the simulations were carried out varying the angles $\xi$ and $\varphi$ from $-2°$ to $2°$, the combinations with less resonant behaviors were found when both relative rotations had the same direction (angles with opposite sign according to Figure 1c).

When the wave crosses an interface with an inclination (water to polystyrene), the change of acoustic impedances provokes more deviations that are reverted to the original inclination when the wave crosses the next interface (polystyrene to culture medium). At certain opposite inclinations of the studied boundaries, the effect of the resonances is still present, for instance, when both relative inclinations are the same ($\xi = varphi$). In this case, the wave returns to the transducer following the same path, which produces constructive interference with the next generation of waves and creates standing waves. This did not occur when both boundaries were relatively inclined in the same direction, because the waves were continuously deviating from the transducer after crossing each interface.

LIPUS heating in the culture medium and the dish was modeled for a geometry with no resonant effect by standing waves, in accordance with the experimental setup ($TD = 10$ mm, $CM = 5$ mm). The temperature increment produced in the dish after 5 min of LIPUS with the largest intensity used in experiments was 0.76 °C at the very center of the well (Figure 4). We can notice the correspondence of this temperature distribution with the acoustic field pattern of Figure 3b. Heat propagation in liquids, although influenced by the thermal conductivity, has a predominant contribution of natural and force convections. In this model, it was omitted the effect of agitation produced by the ultrasound radiation force that could act as an external force convection and should permit a more distributed heat. Although nature convection was included based on [27], the effect of acoustic agitation should be more significant. Therefore, real-temperature increments in the dish produced by ultrasound absorption should be smaller and more uniform than those shown in Figure 4 and can be comparable to the measured increments of 0.3 °C reported by [13].

### 4.5. Study Limitations

One of the limitations of the present study is that cell viability was assessed with only one methodology. Despite the fact that MTT assay is considered the gold standard for cell viability evaluation, it relies on the mitochondrial ability to metabolize the tetrazolium salt into a purple non-soluble formazan precipitate by metabolically active cells [32]. Since many cell culture conditions and treatments may affect mitochondrial activity [33], other methods to corroborate cell viability and proliferation should be considered. Further, a higher number of experimental evaluations would be desirable to increase the statistical power of the present study.

Another limitation that has been noticed is that we cannot rule out that the decrease in cell viability observed with cell passaging after LIPUS treatment assay is a consequence of the well-known LIPUS effect to deform or even disrupt the cell membrane [34]. Thus, it may be possible that cell trypsinization exerted additional stress on the cell population that was unable to repair its membrane. Although we did not observe an immediate negative effect on the cell adhesion capacity in subcultures of LIPUS-treated cells, it may be possible that those cells initiated a subsequent cell death program.

Furthermore, the use of FEM for our models has certain limitations. For instance, acoustic simulations have required to transform 3D time-dependent equations into 2D harmonic solutions after assuming the pressure being separable in spatial and temporal components [21]. However, real acoustic propagation is a rather complex phenomenon with linear and non-linear components that depend on the acoustic intensity and the properties of the propagating media. Moreover, the solver steps used for the time-dependent simulation for heat propagation simplified the required computation by using the calculated harmonic and stable acoustic field as the input of the transient model. The implications of this two-step solution for thermoacoustic simulations were studied and discussed previously in another paper [22].

Additionally, when the ultrasound is emitted at a certain frequency, harmonics are also produced. The effect of these harmonics was not included in this analysis, which can be considered another limitation of this study. However, the used ultrasound generator has a measured power ratio of 25 between the fundamental frequency and the third harmonic. This indicates that for the maximum acoustic intensity used in the experiments, 1.6 W/cm$^2$

at 1 MHz, the transducer would be producing a maximum intensity of 0.064 W/cm$^2$ at 3 MHz if it is assumed 100% of efficiency. After considering more realistic losses (lower efficiency), the intensity of the harmonic signal would be negligible, and we would not expect significant effects in cell proliferation.

## 5. Conclusions

MIO-M1 cells treated with LIPUS displayed higher tolerance to ultrasound when stimulated using the well-on-water-surface setup, when compared with the well-on-transducer configuration using gel as a coupling medium. Results indicated the cell viability was significantly affected by LIPUS only at 1.6 W/cm$^2$ at 20% and 50% DFs using the water-based setup. However, when the cells were subjected to subculturing, significant cell death was observed 48 h after cells passage.

The setup was validated with acoustic field measurements and thermoacoustic models. Modeled heat produced by LIPUS under the largest intensity used in our experiments produced a temperature increase of 0.76 °C at the center of the dish. This temperature should not influence the cell viability. The resonances produced by standing waves in the system were also studied. Including a small 1° inclination of the dish relative to the transducer surface would reduce the effect of resonances and, consequently, would permit to have more reproducible experimental results.

**Author Contributions:** Conceptualization, I.P.-N. and M.I.G.; methodology, I.P.-N. and M.I.G.; software, M.I.G.; validation, I.P.-N. and M.I.G.; formal analysis, I.P.-N., M.I.G. and D.E.M.-S.; investigation, I.P.-N., M.I.G. and D.E.M.-S.; resources, M.I.G., A.O., A.A., J.G.-M., L.L. and A.V.; data curation, I.P.-N., M.I.G. and D.E.M.-S.; I.P.-N., M.I.G., A.O., A.A., J.G.-M and D.E.M.-S.; visualization, I.P.-N. and M.I.G.; supervision, I.P.-N. and M.I.G.; project administration, M.I.G.; funding acquisition, M.I.G., L.L. and A.V. All authors have read and agreed to the published version of the manuscript.

**Funding:** This research was funded by CONACyT, grant number 257966, ERAnet-EMHE, grant number 200022, CYTED-DITECROD, grant number 218RT0545, and AMEXCID-AUCI 2018-2020 project number IV-8.

**Institutional Review Board Statement:** Not applicable.

**Informed Consent Statement:** Not applicable.

**Data Availability Statement:** The data presented in this study are openly available in FigShare at 10.6084/m9.figshare.13489770, reference [35].

**Acknowledgments:** Authors would like to thank Rubén Pérez Valladares and José Hugo Zepeda Peralta for their technical support during acoustic field measurements and experimental setup adaptations.

**Conflicts of Interest:** The authors declare no conflict of interest. The funders had no role in the design of the study; in the collection, analyses, or interpretation of data; in the writing of the manuscript, or in the decision to publish the results.

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
