# Peer review of "Low-Intensity Pulsed Ultrasound Effect on MIO-M1 Cell Viability: Setup Validation and Standing Waves Analysis"

_applsci, doi:10.3390/app11010271_

Round 1
Reviewer 1 Report
The work is well written and try yo put some light on the problems affecting the repeatability of ultrasonic traetment of cell cultures with in vitro experiments.
However I think that the solution proposed by the authors for reducing standing wave effect is a little bit far from a real useful set-up. My experience suggest that filling completely the well containing the cell culture and sealing it with some soft silicone caps ( i.e without air trapped) can be more easy to realize and would it be more appropriate for discriminate from the standing waves effects and the vibrations, as it was already done in the two experimental set-up proposed for ultrasonic treatment.
I think that the authors should investigate also this scheme in order to provide a well defined set-up.
Reviewer 2 Report
See attached document

Reviewer 3 Report
The methodology and results are sound.
Author Response
We are thankful for your comment.